# Liver Transplantation for Biliary Atresia in Adulthood: Single-Centre Surgical Experience

**DOI:** 10.3390/jcm10214969

**Published:** 2021-10-26

**Authors:** Miriam Cortes-Cerisuelo, Christina Boumpoureka, Noel Cassar, Deepak Joshi, Marianne Samyn, Michael Heneghan, Krishna Menon, Andreas Prachalias, Parthi Srinivasan, Wayel Jassem, Hector Vilca-Melendez, Anil Dhawan, Nigel D. Heaton

**Affiliations:** 1Liver Transplant Surgery, Institute of Liver Studies, King’s College Hospital, London WC2R 2LS, UK; m.cortescerisuelo@nhs.net (M.C.-C.); christina.boumpoureka@nhs.net (C.B.); noelcassar@svhg.ie (N.C.); krishna.menon@nhs.net (K.M.); andreas.prachalias@nhs.net (A.P.); parthi.srinivasan@nhs.net (P.S.); wayel.jassem@kcl.ac.uk (W.J.); hector.vilca_melendez@kcl.ac.uk (H.V.-M.); 2Hepatology Department, Institute of Liver Studies, King’s College Hospital, London WC2R 2LS, UK; d.joshi@nhs.net (D.J.); michael.heneghan@nhs.net (M.H.); 3Paediatric Liver, GI and Nutrition Centre and Mowat Labs, King’s College Hospital, London WC2R 2LS, UK; marianne.samyn@nhs.net (M.S.); anil.dhawan@nhs.net (A.D.)

**Keywords:** extra-hepatic biliary atresia, aneurysm, shunts, liver transplantation

## Abstract

Background: Biliary atresia (BA) is the most common indicator for liver transplant (LT) in children, however, approximately 22% will reach adulthood with their native liver, and of these, half will require transplantation later in life. The aim of this study was to analyse the surgical challenges and outcomes of patients with BA undergoing LT in adulthood. Methods: Patients with BA requiring LT at the age of 16 or older in our unit between 1989 and 2020 were included. Pretransplant, perioperative variables and outcomes were analysed. Pretransplant imaging was reviewed to assess liver appearance, spontaneous visceral portosystemic shunting (SPSS), splenomegaly, splenic artery (SA) size, and aneurysms. Results: Thirty-four patients who underwent LT for BA fulfilled the inclusion criteria, at a median age of 24 years. The main indicators for LT were synthetic failure and recurrent cholangitis. In total, 57.6% had significant enlargement of the SA, 21% had multiple SA aneurysm, and SPSS was present in 72.7% of the patients. Graft and patient survival at 1, 5, and 10 years was 97.1%, 91.2%, 91.2% and 100%, 94%, 94%, respectively Conclusions: Good outcomes after LT for BA in young patients can be achieved with careful donor selection and surgery to minimise the risk of complications. Identification of anatomical variants and shunting are helpful in guiding attitude at the time of transplant.

## 1. Introduction

Biliary atresia (BA) is the most common indicator for liver transplant (LT) in children. It is a progressive obliterative cholangiopathy of unknown cause with an incidence of 1:6000–20,000 live births, and, untreated, results in death within the first years of life without Kasai portoenterostomy (KPE) [1,2,3]. KPE was introduced in the late 1950s and significantly improved early outcomes of BA, especially if performed within the first 90 days of life [4,5]. In total, 55–60% of infants undergoing KPE during the first 3 months of life clear their jaundice, however two thirds will require liver transplant within 10 years or more, as they will develop liver fibrosis and cirrhosis and portal hypertension (PHT) with synthetic failure, variceal bleeding, and recurrent cholangitis. Those patients who achieve good biliary drainage after KPE reach adolescence with their native liver intact, however, some have clinically evident PHT and come to LT subsequently [6,7].

As a consequence, there is an emerging population in the adult hepatology services of young people (≥16 years old) with BA who will need LT who differ significantly from the general adult population requiring a LT [8,9]. The timing of transplant listing can be difficult and prognostic scores, such as the Model for End stage Liver Disease (MELD), are less helpful except after decompensation [10]. The most common indications for LT in these patients include refractory variceal bleeding, recurrent cholangitis, chronic cholestasis with synthetic failure and loss of weight and muscle bulk, hepatopulmonary syndrome, portopulmonary hypertension, and hepatocellular carcinoma (HCC) [8].

These patients also differ in terms of their disease profile, which is dominated by PHT and the consequences of their collateral circulation, which pose surgical challenges and can cause postoperative complications. Uchida et al. reported that living donor liver transplant (LDLT) in adult patients with BA had higher rates of post-transplant intestinal perforation, intra-abdominal bleeding requiring relaparotomy, and biliary leakage, as well as lower survival in comparison to children [9]. Adult patients with BA who undergo LT face additional challenges, including optimal timing of transplant, waiting list mortality, complexity of surgery, and higher risk of complications [11,12]. Superina summarised the variety of surgical findings and management during LT in children with BA, but little has been reported in adults [13]. The aim of this study was to review our experience of adult BA patients undergoing LT.

## 2. Materials and Methods

### 2.1. Study Population

This was a retrospective, single-centre review of prospectively collected data of patients with BA requiring LT, age ≥ 16, in our unit between January 1988 and June 2020. Inclusion on the waiting list for LT was in accordance with MELD and United Kingdom Model for End-Stage Liver Disease scores after transplant assessment and patient consent was obtained [14,15,16]. Patients not meeting minimum listing criteria or with variant syndromes were listed by national appeal. Patients were supported by a multidisciplinary team providing care for young adults (16 to 25 years).

### 2.2. Recipient Characteristics

Demographic and clinical data, including pretransplant status, perioperative variables, and short and long-term outcomes were collected. Immunosuppression was with tacrolimus and steroids. Prior to 1994, patients received cyclosporin, azathioprine, and prednisolone. Synthetic failure was defined as serum albumin < 30 g/L and the presence of ascites clinically and on ultrasound. Refractory cholangitis was defined as failure of clearance of jaundice following appropriate treatment of cholangitis with antibiotics and recurrence of cholangitis within 6 months. Hepatopulmonary syndrome was diagnosed by nuclear medicine scan. No cases of pulmonary hypertension were detected.

### 2.3. Imaging Assessment

Pre- and post-transplant imaging, including computed tomography (CT) and liver ultrasound (US), were reviewed to assess liver morphology, splenic artery (SA) diameter, presence of visceral aneurysms, portal vein flow, and existence and pattern of portosystemic shunting. All patients had at least one CT before transplant and liver US was repeated every 6 months while on the waiting list. The SA was considered enlarged when it measured >4 mm or was 150% of the size of the common hepatic artery (HA) on CT [17]. The presence of spontaneous portosystemic shunts (sPSSs) was determined by the presence of varices on CT measuring ≥5 mm in diameter. Four main territories of sPSSs were considered: left gastric vein, perisplenic, splenorenal, and retroperitoneal [18]. The recommendation was to tie the shunts noted at listing; however, the final decision was made by the transplant surgeon during surgery. SPSSs were closed either by ligation or transfixion. Small varices were treated with Argon coagulation.

### 2.4. Postoperative Care and Follow-Up

Graft function was classified according to Olthoff et al.’s criteria with ‘early allograft dysfunction’ (EAD) defined as the presence of one or more of the following criteria: serum bilirubin ≥ 10 mg/dL and international normalized ratio ≥ 1.6 on day 7; alanine or aspartate aminotransferase > 2000 IU/L within the first 7 days after the transplant [19]. Follow-up was up to 1 November 2020. Last follow up was defined as the last documented clinical visit at the time of data collection. This study was approved by the Hospital Institutional Review Board.

### 2.5. Statistical Analysis

Continuous variables were represented as median and range and compared using Wilcoxon sum rank test, categorical variables were represented as absolute number and percentage. Overall patient and graft survival were determined using the Kaplan-Meier method. A *p* value < 0.05 was considered significant. All analyses were performed using SPSS (IBM, Armonk, 150 New York, version 24).

## 3. Results

### 3.1. Recipients

Five hundred and one patients with BA underwent LT in our centre between October 1989 and June 2020, of whom 34 were ≥16 years old. Median age at transplant was 24 years old (16 to 44), and 20 were female (58.8%). All patients underwent KPE at a median age of 7 weeks (3–21), and three had additional pretransplant surgical procedures, including splenectomy, Roux-en-Y refashioning, and duodenostomy reversal after a barrel duodenostomy was performed during KPE due to injury. The most common indications for transplant were synthetic failure (32.4%), recurrent cholangitis (29.4%), and progressive jaundice (17.6%). Less frequent indications were fatigue and lethargy (two), acute severe refractory variceal bleeding (three), hepatopulmonary syndrome (one), and suspected HCC (one) (Table 1). Four patients had overt hepatic encephalopathy (HE), three had synthetic failure and one had severe PHT. Serum ammonia levels were available in six patients and were elevated in five, ranging from 82 to 300 µmol/L.

MELD score increased significantly by the time of transplant when compared to listing (15 versus 12 (*p* < 0.0005)) and median waiting list time was 292 days (15–2425). Two patients were admitted to hospital for sepsis prior to transplant, and eight were prioritized due to progressive deterioration from sepsis, weight loss, and recurrent bleeding.

### 3.2. Donor

All grafts were from deceased donors except for one right lobe liver from a living donor. There were two from donors after circulatory death (DCD), used as whole livers; and 31 donors after brain death (DBD), used as whole grafts in 21 cases and partial grafts in 10 cases: seven split right lobe (RL), two reduced RL, and one reduced left lobe. Median donor age was 39 years (14–76). Both DCD donors were <40 years old with a functional warm ischemia time (defined as time from systolic blood pressure <50 mmHg and/or oxygen saturation < 70% until cross-clamp) of less than 20 min (Table 2).

### 3.3. Imaging Findings

Imaging prior to transplant was available in all but the first six patients in our series. The liver morphology on CT and/or magnetic resonance imaging were characterized by significant central hypertrophy (segment 4 and 1) with atrophy of the right lobe and the left lateral segment (Figure 1A). Nineteen patients (57.6%) were found to have significant SA enlargement (Table 3). Of these, seven (21.2%) had multiple splenic artery aneurysm (SAA) measuring up to 53 mm. In two patients (6%), there were aneurysms in the common and right HA, respectively (Figure 1B,C).

In total, 24 (72.7%) patients were found to have significant intra-abdominal variceal collaterals distributed in different territories (Figure 2A–D); retroperitoneal varices were most frequently observed (22 cases; 64.7%), followed by perisplenic (41.2%), lienorenal (26.5%), and left gastric territory (seven cases; 20.6%) (Table 3). These shunts appear to contribute to abnormalities observed in the portal vein (PV), as detected on imaging. On CT, 16 (47%) patients had an attenuated PV, of whom 14 (87.5%) had SPSS compared to 7 (39.9%) of 18 (53%) patients with a normal PV appearance on CT. The shunts in patients with attenuated PV tended to have more territories involved, with a median of two territories (1–4) compared to one (1–3) in the normal PV group, but this was not statistically different. On the USs, 14 patients (41.7%) either had absent flow (five), reversed flow (five), or low-velocity antegrade flow (two), of which 11 (78.5%) had attenuated PV on CT. Portal vein thrombosis (PVT) was found in one patient at transplant and was managed with eversion thrombectomy. This patient had an attenuated PV on CT and reversed flow on their US. In six patients, the PV was reported as small in the operating notes, one patient required an inter-positional graft, and another had a low anastomosis after resecting the narrowed extrahepatic PV segment.

Two patients had syndromic BA, one with complete situs inversus with absent retrohepatic vena cava and another had an absent inferior vena cava, replaced HA from the left gastric artery, lienorenal, left gastric and perisplenic shunts, and a portosystemic shunt between left PV and the inferior vena cava.

### 3.4. Surgical Events

The median surgical time was 460 (230–765) min and median recipient hepatectomy time was 160 (90–280) min, with a median blood loss of 5 litres (1–16.7) (Table 2). Piggy-back was the preferred technique in 25 transplants (75%) with temporary portocaval shunt depending on the intraoperative bleeding and degree of venous shunting. In six patients, the Roux loop was lengthened because it was considered short (<40 cm). On six occasions, the SA was ligated because of the existence of SAA. SPSS closure was only described in the operation notes of five patients.

During surgery, intraoperative bowel injury was reported in three patients (8.8%): in one, the Roux loop had fistulated into the duodenum requiring repair; two had multiple serosal tears in the small and large bowel, and one required resection of 70 cm of small bowel. One patient with situs inversus had an upper pole partial splenectomy to make room for a split right lobe graft.

### 3.5. Postoperative Course

The median intensive care unit stay was 2 days (1–12) and median hospital stay was 15.5 days (9–46). EAD was present in six patients (17.6%); 12 patients had early acute cellular rejection (35.2%) diagnosed by transaminitis, confirmed histologically and treated with steroids pulse (Table 4).

Three patients required emergency surgery during the first 14 days after transplantation for colonic perforation and ileostomy formation on day 9, bleeding on day 10, and wound dehiscence with muscle necrosis on day 15, respectively. There were no early or late vascular complications except for one patient who presented with left lateral segment ischaemia 20 days after surgery, related to left accessory HA injury during the procurement that resolved with antibiotics. Biliary complications occurred in seven patients (20.5%)—three cut surface leaks and four anastomotic biliary strictures, one of which required surgical revision. Five patients also had wound infection.

Liver function at 4-year follow up was normal in all patients except six with raised alkaline phosphatase and/or gamma-glutamyl transpeptidase. In two of these patients, liver biopsy revealed bridging fibrosis and nodular transformation, and the other patient had an anastomotic biliary stricture treated by percutaneous transhepatic dilatation.

The median follow-up was 4 years, with a maximum of 22 years. The first patient in our series died at 2 years from lymphoma, and another one died of multiorgan failure before re-transplantation in the context of poor compliance and bridging fibrosis with nodular transformation within the graft; one patient required re-transplantation 11 months after the first transplant for ductopenic rejection. Graft and patient survival at 1, 5, and 10 years was 97.1%, 91.2%, 91.2% and 100%, 94%, 94%, respectively.

## 4. Discussion

BA is the most common indication for LT in children and is associated with excellent outcomes [20]. Early diagnosis of BA, early KPE, refinements of surgical technique, and centralization of care have contributed to increased survival with the native liver intact and delaying LT to later in life [1,21,22]. Up to 44% of the children with BA post-KPE will reach adulthood with their native liver intact; of these, approximately half will require a LT later [23,24]. The small number of patients requiring LT for BA in adulthood has not allowed for standardization of their management and issues such as timing of listing, surgical challenges, and outcomes have not been fully characterised.

Factors predicting outcome after KPE include age at Kasai, degree of inflammation in the liver biopsy, presence of ascites, nodular liver, presence of polysplenia, and postoperative jaundice resolution [13,25]. As these children grow with their native liver, they usually have well-preserved synthetic function, but with clinical evidence of PHT. Scoring systems, such as MELD, underestimate disease severity until the liver decompensates. The creation of a scoring system that assists clinicians in deciding when to list has been attempted. Jain et al. identified serum bilirubin ≥ 21 umol/L, low serum creatinine, the presence of PHT and variceal bleeding at 16 years, and cholangitis in adolescence as predictors of future need for LT in young BA patients [26]. The development of a specialist team to care for young patients with liver disease transitioning to adult services has emerged over the last decade, especially in centres offering both paediatric and adult liver transplantation [27]. This team advocates for these patients as they face challenges with regards to listing, organ allocation, and prolonged waiting times for suitable grafts, which can compromise their outcome [8,26].

The median waiting list time was 292 days, which was longer than that for children with BA in our centre, for whom the wait time was 177 days (121–243), and also for 3008 adult patients with chronic liver disease who waited 110 days for transplants over this time period [8]. Contributing factors appear to be the need for a good-quality liver graft to minimise the risk of graft failure and because the hepatectomy time may be longer due to previous surgery and PHT, syndromic BA, and/or malrotation. Additional procedures may be necessary, including shunt ligation to optimise the portal flow, PV reconstruction, and ligation of the SA if SAA or gross splenomegaly are present. The median hepatectomy time in our series was 160 min, with the longest at 280 min, which is above the 60–120 min considered as the standard. Ausania et al. defined a cut-off of 180 min for prolonged hepatectomy, which was based on with previous abdominal surgeries, surgical experience, and PHT [28]. In total, 44% of our patients fulfilled these criteria. Others have shown that young adults with chronic liver disease wait longer on the list, have higher mortality whilst waiting, and are less likely to be re-transplanted [29]. Greater understanding of these issues is important to enable equality of access to transplantation and re-transplantation.

Finding a size match organ can be a challenge, as these patients are young and usually thin with narrow abdomens. The median BMI at listing was 22.6 and 55.8% were women, who often have a smaller anterior–posterior diameter, requiring a smaller liver. No difference was found in waiting time according to type of donor (DBD/DCD/LD) or type of graft (whole/split/reduced). A right lobe split is usually a good-sized graft for young adults, however, the main limitation in our centre is the prolonged cold ischaemia time, as the transplant surgery will often happen sequentially after the LLS has been implanted in another child. Ideally, the transplant should be performed simultaneously. In our series, a refashioned Roux loop was performed in five cases when it was <40 cm. However, if the surgery was complex and associated with significant blood loss, the Roux loop was not extended but utilised to avoid further dissection. No revision surgery has been required to date for ascending cholangitis in these patients.

Insufficient portal flow early after liver transplant because of spontaneous portosystemic shunts can be associated with graft dysfunction, PVT, or HE [30,31]. It has been considered that if the portal flow is good during the transplant, there is no need to ligate the collaterals, however, there is increasing evidence that not doing so can compromise long-term outcomes. Gomez-Gavara et al. reported the outcomes in 79 patients with sPSS, in whom the decision to ligate shunts during transplant was made by two senior surgeons based on the effect on the portal flow after shunt clamping. In those patients that did not have shunts ligated, they observed increased rates of low-grade encephalopathy, PVT, and other postoperative complications, as well as lower patient and graft survival at 1, 2 and 4 years [32]. In children with biliary atresia, it is recommended to tie all shunts to optimise portal flow and decrease the risks of insufficient portal flow and PVT after transplant; the same approach needs to be maintained in patients with BA being transplanted in adulthood [33].

The incidence of SPSS in our series was 68%, which was above the incidence described in adult patients with end-stage liver disease, reported at between 20 and 40% [32,34,35]. The rationale behind this is that severe, long-standing PHT present in these patients as a consequence of the fibrosis and remodelling of the liver leads to increased portosystemic shunts and decreased portal flow with hypoplastic or small portal veins. This phenomena, described by Kasai, was considered to be secondary to a decreased number of intrahepatic portal vein radicals in an inflamed and scarred liver, however, it was thought to reverse or improve with successful KPE [36]. In our series, 16 patients (47%) had attenuated PV on the pretransplant CT and the US showed abnormal portal flow in 14 cases (41.7%) as a reflection of long-standing PHT and development of SPSS.

SPSS ligation was reported in the operation notes of five patients and it is likely that more shunts were tied. Increasingly, our approach has been to tie all shunts during surgery. Kashara et al. reported fastidious division of all collaterals routinely in children with BA at transplant [37]. Some patients require PV surgery using either an interpositional vein graft as replacement or PV plasty of the anterior wall with a strip of vein obtained either from the donor or recipient’s native right or left PV. The hypoplastic segment is usually the extrahepatic portion of the PV and becomes the normal size as the bifurcation of the SMV and splenic vein is approached [38,39].

The incidence of SAA in patients with cirrhosis is in the range of 7–17% and has been positively related to the size of the SA and to long-standing PHT with significant and large portosystemic shunts and splenomegaly [40,41,42]. This is in agreement with the incidence of SAA in our series of 21.2% (seven patients). All of our cases were multiple; SAAs are described as multiple in 50% of cases in the published literature [40]. SA ligation was performed in six cases during transplant and the seventh patient underwent radiological embolization after LT. No spontaneous SA rupture was encountered. Two patients presented with HA aneurysm (HAA) on imaging during transplant assessment; the oldest patient in our series at the time of transplant (44 years old) had a calcified aneurysm in the common HA and the second patient had an aneurysm in the right HA. Both were close to the bifurcation of the right and left HA. Visceral aneurysms have been reported in association with chronic liver disease rather than atherosclerosis. The incidence of visceral aneurysms in the literature is low 0.01–0.2% in the general population and relates to atherosclerosis, trauma, or vasculitis. HAAs are likely related to the increased arterial flow associated with progressive liver disease, hyperdynamic state, and the effect of systemic vasodilatation with loss of vascular resistance.

SA ligation is useful to manage SAA and to decrease the risk of SA steal syndrome (SASS), defined as liver hypoperfusion secondary to a decrease in HA flow in the absence of HAS or HAT, which is associated with increased blood flow through an enlarged SA [43]. Some authors have reported that SA greater than 4 mm or greater than 150% the HA size is associated with SASS [44]. The findings on the USs with SASS indicate a high-resistance HA waveform with low diastolic or reversal of diastolic flow and a resistance index (RI) greater than 0.8; however, the final confirmation is by CT angiography [17]. Reported complications of SAS have included early graft dysfunction, biliary ischemia, and cholangiopathy, with reports of re-transplantation [43]. In our series, 18 patients (52.9%) had an enlarged SA > 4 mm on the pretransplant CT, all except one of which were 150% of the HA diameter. None of the postoperative USs showed an elevated RI. Our practice has changed, especially with regards to ligation of the SA if it is significantly enlarged with marked splenomegaly. We have encountered patients with attenuated HA on CT months after liver transplant with persistent hypersplenism. Aside from the partial splenectomy in a patient with a situs inversus in this series, we have performed a partial splenectomy in three other patients not included in this series for portal hypertension and massive splenomegaly at transplant. The role of partial splenectomy in conjunction with ligation of collaterals may give better long-term outcomes by reducing SA flow and splenic bed venous pressure in conjunction with collateral ligation. Partial splenectomy has been described in patients with haematological conditions, tumours, or trauma, or to decrease the degree of portal hypertension and small-for-size syndrome in living donor liver transplant [45,46]. Nam et al. proposed performing splenectomy or splenic artery ligation when the spleen volume with respect to the body surface area exceeded a certain threshold [47].

As reported in children being transplanted for BA post-KPE, bleeding and bowel perforation are common complications [9,48]. Yanagi et al. reported an incidence of perforation in 24% of young adults after LT for BA at a median of 9 days [48]. Contributing factors include severe surgical adhesions, thermal injury during dissection and haemostasis, PHT and bowel congestion, and impaired healing associated with immunosuppression [49]. Segura Sampedro et al. reported five adult patients requiring LT for BA, while two patients developed wound dehiscence, bowel perforation, and acute rejection [50]. Uchida et al. reported lower patient and graft survival in 47 adult patients with BA undergoing LDLT when compared to children and identified previous abdominal surgeries, MELD score, hepatopulmonary syndrome, and laparotomies after LT as complications that are independent risk factors for poor survival.

The overall survival in our series was similar to that reported by Yanagi et al. in 24 patients over 13 years of age undergoing LDLT for BA and better than those reported by Uchida in LDLT; however, the patients in Uchida’s series had more surgeries before transplant and included patients receiving incompatible blood group grafts and had hepato-pulmonary syndrome.

It is important to highlight that very few patients underwent formal neurocognitive assessment unless HE was overtly suspected; however, there is increasing evidence that these patients have subclinical HE, which relates to the size and number of SPSSs, and if these are not closed, hyperammonaemia may persist after LT despite normal liver function. These patients tend to have lower platelet counts and plasma ammonia levels may be elevated to surprising levels. Platelet count may potentially be used as a surrogate marker for effective management of SASS and splenomegaly post-LT. The screening of HE before transplant is important as it will affect the development and quality of life of the patient and could be used to expedite inclusion on the wait list and highlight the need for intraoperative collateral ligation [8].

The main limitation of this paper is the retrospective nature. To our knowledge, this is the largest study of adult patients undergoing LT for BA with deceased donors, exploring the perioperative and surgical variables. To achieve good outcomes in this population, patients need to be listed early with a focus on PHT and blood ammonia levels; to receive a good quality graft, with a surgical plan to deal with the PV and venous collaterals, SAA, and massive splenomegaly; and to receive follow up by a specialist team to minimise the risk of graft loss related to poor compliance.

## Figures and Tables

**Figure 1 jcm-10-04969-f001:**
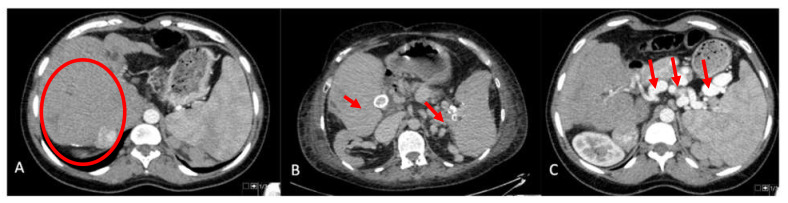
CT imaging pretransplant. (**A**) Central hypertrophy of the liver with atrophy of the left lateral segment and right lobe. (**B**) Calcified hepatic artery and splenic artery aneurysms. (**C**) Significant enlargement of the splenic artery and multiple aneurysms.

**Figure 2 jcm-10-04969-f002:**
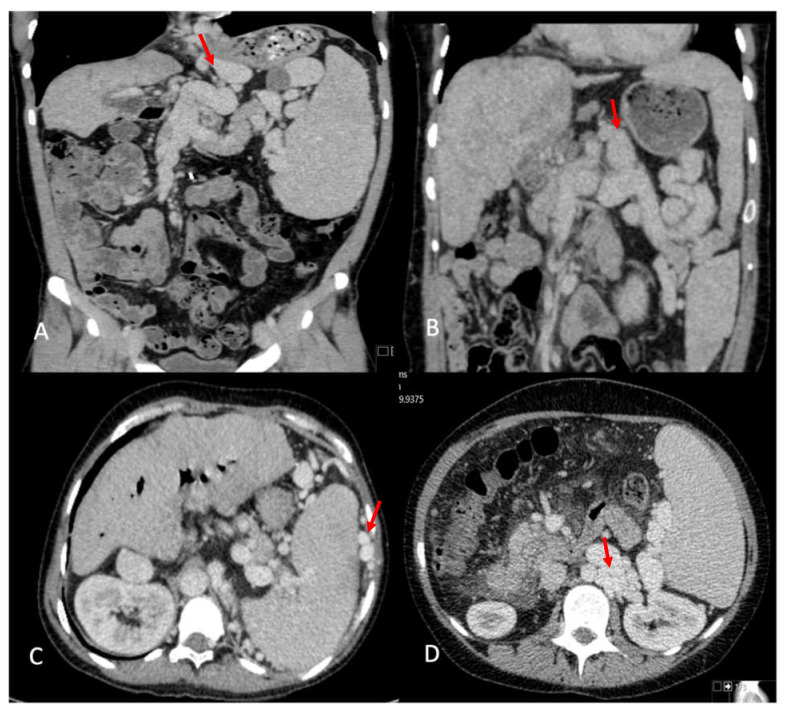
Visceral portosystemic shunts on imaging. (**A**) Left gastric vein varix. (**B**) Lieno-renal shunt. (**C**) Perisplenic varices. (**D**) Retroperitoneal varices.

**Table 1 jcm-10-04969-t001:** Recipient characteristics.

Variables	Results	
Age (years)	24 (16–44)	
BMI	22.8 (19–39.1)	
Gender (female)	19 (57.6%)	
Age at KPE (weeks)	7 (3–21)	
Time on the waiting list (days)	292 (15–2425)	
MELD score *		*p* < 0.005
Listing	12 (6–25)	
Pretransplant	16 (6–33)	
Indications for transplant		
Synthetic failure	11 (32.4%)	
Recurrent cholangitis	10 (29.4%)	
Progressive jaundice	6 (17.6%)	
Fatigue and lethargy	2 (5.9%)	
Acute bleeding	3 (8.8%)	
HPS	1 (2.9%)	
Suspected HCC	1 (2.9%)	

BMI, body mass index; HCC, hepatocellular carcinoma; HPS, hepatopulmonary syndrome; KPE, Kasai portoenterostomy; MELD, model for end-stage liver disease. Continuous variables are expressed as median and range except for MELD score, which is expressed as mean and standard deviation. Categorical variables are expressed as number and percentage. * Wilcoxon signed-rank test (significant *p* < 0.05).

**Table 2 jcm-10-04969-t002:** Donor characteristics and surgical variables.

Variables	Results
Type of donor	
DBD	31 (91%)
DCD	2 (6%)
LD	1 (3%)
Type of deceased graft	
Whole liver	23 (70%)
Split RL	7 (21%)
Reduced RL	2 (6%)
Reduced LL	1 (3%)
CIT (min)	585 (92–960)
Surgical time (min)	455 (230–765)
Hepatectomy time (min)	165 (90–280)
Surgical technique	
Piggy-back	25 (76%)
Cava replacement	8 (25%)
Blood loss (litres)	5 (1–16.7)

DBD, donors after brain dead; DCD, donors after circulatory dead; CIT, cold ischemia time; LD, living donor; LL, left lobe; RL, right lobe. Continuous variables are expressed as median and range. Categorical variables are expressed as number and percentage.

**Table 3 jcm-10-04969-t003:** Findings on imaging.

Findings	Results
Arterial aneurysm	
Splenic artery	7 (21.2%)
Hepatic artery	2 (6%)
Enlargement of the splenic artery	18 (52.9%)
Intrabdominal variceal shunting	
Total number of patients	24 (68.6%)
Retroperitoneal	22 (64.7%)
Perisplenic	14 (41.2%)
Lieno-renal	9 (26.5%)
Left gastric territory	7 (20.6%)
Portal vein findings on US	
Low velocity antegrade	2 (6.1%)
Retrograde	5 (15.2%)
No flow	5 (15.2%)

US, ultrasound (the size of the spleen was measured via US, the presence of aneurysm, size of the splenic artery, and presence of variceal shunting were described according to the computed tomography). Continuous variables are expressed as mean ad standard deviation. Categorical variables are expressed as number and percentage. Wilcoxon signed-rank test (significant *p* < 0.05).

**Table 4 jcm-10-04969-t004:** Postoperative course.

Variables	Results
ICU stay	2 (1–12)
Hospital stay	15.5 (9–46)
EAD	6 (17.6%)
Early cellular rejection	12 (35.2%)
Emergency surgery after LT	3 (8.8%)
Biliary complications	
Cut surface bile leak	3 (8.8%)
Biliary stricture	4 (11.7%)
Dead	1 (3%)
Re-transplantation	1 (3%)

EAD, early allograft dysfunction according to Olthoff’s classification; ICU, intensive care unit; LT, liver transplantation. Continuous variables are expressed as median and range. Categorical variables are expressed as number and percentage.

## Data Availability

There is no supporting data.

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
