# Peer review of "Liver Transplantation for Biliary Atresia in Adulthood: Single-Centre Surgical Experience"

_jcm, 2021, doi:10.3390/jcm10214969_

Round 1

Reviewer 1 Report

The authors investigated the surgical outcomes of patients with Biliary Atresia (BA) undergoing Liver Transplantation (LT) in adulthood.

They referred in detail the efficacy of LT for BA in young ages.

Therefore, I will be able to recommend it for acceptance with several points.

It is understandable that a LT for BA at a younger age, before the liver and surrounding conditions deteriorate, may be beneficial. So at what stage should we begin to consider LT? It would be better to mention whether if an algorithm or scoring system can be constructed.

Author Response

Dear Reviewer,

Thank you very much for your comments.  We wanted to focus this paper from a surgical perspective and the importance of having a surgical plan to deal with the porto-systemic shunts and arterial aneurysm to improve long-term outcomes. I agree that  a reliable algorithm to help clinicians decide when to transplant this patients or to prioritise this patients on the waiting list is paramount; my colleagues from King's recently publish a paper  (reference 26 in the paper) where they report that elevated bilirubin and low creatinine was associated with the need of a LT for EHBA after the age of 16 years-old. Recurrent cholangitis during adolescence and significant portal hypertension by 16 years  were also associated to 7-8 fold increase risk for having a LT.  I agree this is a very important topic and further and more accurate evidence is needed. 

Reviewer 2 Report

Thank you for giving me the opportunity to review this paper from an accomplished team in the UK.  The authors present their series of LT for BA in adult populations. 

There are a few considerations;
1. In this study, the authors collected the data of patients ≥ 16-year-old. How did the authors decide the cut-off of the age?

2. The authors described the change of spleen size before and after LT. However, some patients underwent splenic ligation, splenectomy, or SPE.  The improvement of spleen size and platelet count after LT is well known. It would not be necessary to mention about spleen in this study. 

3. In all figures, the authors should put the arrows pointing to each anatomical site. It can be helpful for readers.

Author Response

Dear author,

Thank you very much for your comments. In UK, and in the clinical setting, the cut-off age to transfer patient's to the adult services is 16, hence, the decision to choose this cut-off value. When patients are ≥16 years-old, the transplant is performed by the adult surgeons which are not always familiarised with the differences in patients with paediatric liver diseases needing a transplant in adulthood.

I agree that the improvement of spleen size and platelet count after LT is well known, and it can be influenced by other factors. This information has been removed from the methodology and the results. 

I have added to all figure arrows pointing out the subject matter. I have attached the paper with the changes tracked.